# Creation and Validation of a Portuguese Version of the UCLA Scleroderma Clinical Trial Consortium Gastrointestinal Tract Instrument

**DOI:** 10.3390/ijerph20021553

**Published:** 2023-01-14

**Authors:** Pedro L. Ferreira, Inês Genrinho, Tânia Santiago, Adriana Carones, Carolina Mazeda, Anabela Barcelos, Tiago Beirão, Flávio Costa, Inês Santos, Maura Couto, Maria Rato, Georgina Terroso, Paulo Monteiro

**Affiliations:** 1Centre for Health Studies and Research, Faculty of Economics, University of Coimbra, 3004-512 Coimbra, Portugal; 2Rheumatology Department, Tondela Viseu Hospital Centre, 3504-509 Viseu, Portugal; 3Rheumatology Department, Baixo Vouga Hospital Centre, 3810-164 Aveiro, Portugal; 4Rheumatology Department, Coimbra and University Health Center, 3004-512 Coimbra, Portugal; 5Baixo Vouga Hospital Centre, Rheumatology Department, Egas Moniz Health Alliance, 3810-164 Aveiro, Portugal; 6EpiDoC Unit, CEDOC, NOVA Medical School, NOVA University of Lisbon, 1169-056 Lisbon, Portugal; 7Comprehensive Health Research Center (CHRC), NOVA University of Lisbon, 1169-056 Lisbon, Portugal; 8Rheumatology Department, Vila Nova de Gaia/Espinho Hospital Centre, 4434-502 Vila Nova de Gaia, Portugal; 9Rheumatology Department, Tondela Viseu Hospital Centre Hospital Centre, 3460-525 Tondela, Portugal; 10Rheumatology Department, São João University Hospital Centre Hospital Centre, 4200-319 Porto, Portugal

**Keywords:** systemic sclerosis, gastrointestinal tract, UCLA GIT 2.0, quality of life

## Abstract

(1) Background: The UCLA GIT 2.0 questionnaire has been recognized as a feasible and reliable instrument to assess gastrointestinal (GI) symptoms in systemic sclerosis (SSc) patients and their impact on quality of life. The aim of this study was to create and validate UCLA GIT 2.0 for Portuguese patients with SSc. (2) Methods: A multi-center study was conducted enrolling SSc patients. UCLA GIT 2.0 was validated in Portuguese using reliability (internal consistency, item –total correlation, and reproducibility) and validity (content, construct, and criterion) tests. Criterion tests included EQ-5D and SF-36v2. Social–demographic and clinical data were collected. (3) Results: 102 SSc patients were included, 82.4% of them female, and with a mean sample age of 57.0 ± 12.5 years old. The limited form of SSc was present in 62% of the patients and 56.9% had fewer than five years of disease duration. Almost 60% presented with SSc-GI involvement with a negative impact on quality of life. The means for SF-36v2 were 39.3 ± 10.3 in the physical component summary and 47.5 ± 12.1 in the mental component summary. Total GI score, reported as mild in 57.8% of the patients, was highly reliable (ICC = 0.912) and the Cronbach’s alpha was 0.954. There was a high correlation between the total GI score and EQ-5D-5L and SF-36v2 scores. (4) Conclusion: The Portuguese version of UCLA GIT 2.0 showed good psychometric properties and can be used in research and clinical practice.

## 1. Introduction

Systemic sclerosis (SSc) is a rare multisystem connective tissue disorder of unknown etiology, characterized by excessive collagen deposition in the skin and other organs. It affects predominantly women, with a female-to-male ratio of 3–8:1, and a frequency peak between 45 and 64 years old [1,2]. The annual incidence in Europe is about 19–43/million/year and prevalence is close to 300/million patients, with Norway, France, Croatia, and Greece reporting lower values [3].

Gastrointestinal (GI) tract involvement in SSc has been estimated to be approximately 70–90%. [4,5] However, only 8% of patients appear with severe involvement leading to increased morbimortality [6]. Any region of the GI tract may be involved, with a substantial variability in extension, severity, and disease course. In nearly 10% of cases, GI symptoms could be the first manifestation of SSc [7,8]. Medications, GI microbiota, diet, and other comorbidities are also additional factors that may predispose or worsen GI symptoms [7]. Despite numerous studies having demonstrated that SSc-GI involvement has a significative negative impact on quality of life [7,9], in clinical practice, assessment of skin, and cardiorespiratory and renal involvement, remain the focus of clinical evaluation. 

The original UCLA GIT 2.0 questionnaire was developed in 2009, and since then it has been recognized as a feasible and reliable instrument to evaluate GI symptoms, and to assess its impact on HRQoL in SSc patients [10]. This questionnaire had been translated into and validated in several languages [11,12,13,14,15,16], but not yet into European Portuguese.

Thus, the purpose of this study was to create and validate the UCLA GIT 2.0 (University of California, Los Angeles Scleroderma Clinical Trial Consortium Gastrointestinal Tract) for Portuguese patients with SSc-GI involvement.

## 2. Materials and Methods

We followed good practice principles to translate and culturally adapt health outcomes instruments [17] and the Cosmin taxonomy to validate the obtained Portuguese version [18].

### 2.1. Cultural Adaptation

Before initiating this study, we contacted Dr Dinesh Khanna, the main author of the UCLA GIT 2.0, in order to obtain his permission to validate a Portuguese version of this instrument. We received the information that a Portuguese non-validated version already existed, following the Food and Drug Administration (FDA) guidance on translation [19], and that we should use the version on the University of Michigan’s website.

However, to content validate this Portuguese version, we still felt the necessity to perform a clinical review with two rheumatologists and a cognitive debriefing with patients.

### 2.2. Participants

To validate the UCLA GIT 2.0, we created a questionnaire and invited consecutive patients from five Portuguese hospital centers to participate in the present study between January and April 2022. Inclusion criteria were any one of the following: (1) fulfilment of the 2013 American College of Rheumatology (ACR)/European League Against Rheumatism (EULAR) criteria for the classification of diffuse SSc (dSSc) or limited SSc (lSSc) [20]; (2) the combined EUSTAR (European Scleroderma Trial and Research Group) criteria for very early diagnosis of SSc (VEDOSS) [21]; or (3) presentation of SSc sine scleroderma. These patients were supposed to be autonomous, aged between 18 and 80 years, and with the ability to understand Portuguese and to grant informed consent. Pregnant women were excluded.

A smaller group of patients was randomly selected to complete, a second time, the UCLA GIT 2.0, one month after the previous consultation.

The study was approved by the Ethics Committee of the Regional Health Authority of the Center (ARSC 14/2020) and by the Ethics Committee from one of the hospital centers (CHTV 05/16/09/2021). We also obtained authorization from all the heads of the five rheumatology departments involved. Each participant signed a written consent form before filling in the questionnaire.

### 2.3. Measurement Instruments

Participants in this study were asked to complete sociodemographic, lifestyle, and clinical information, as well as the Portuguese versions of the health status (SF-36v2) and quality of life (EQ-5D-5L) questionnaires.

Regarding the sociodemographic variables, we collected data on sex, age, marital and employment status, and years of education. Smoking and alcohol use were used as proxies for lifestyle variables. Lastly, the clinical variables measured were the SSc subset classification, disease duration since diagnosis, 2013 ACR/EULAR classification criteria, organ involvement, use of pharmacotherapy for the GI system, and complementary diagnostic tests.

The 34-item UCLA GIT 2.0, originally developed in English by Khanna et al., represents one of the few valid and reliable patient-reported outcome (PRO) measures for GI assessment in SSc [10]. It is an improved and shorter version of the 52-item SSC-GIT 1.0 (Scleroderma Gastrointestinal Tract Involvement 1.0) from the same author. It measures eight health-related quality of life (HRQoL) dimensions (reflux, distention/bloating, fecal soilage, diarrhea, social functioning, emotional wellbeing, and constipation) and has been used in several clinical trials of GI treatments in patients with SSc as an outcome measure [22,23]. The total UCLA GIT score is calculated by averaging all the subscales, except the one for constipation, and ranges from 0 (best HRQoL) to 2.83 (worst HRQoL). Its clinically important difference has previously been determined [24] and it has been culturally translated and validated in different languages [11,12,13,14,15,16,25]. The levels of GI severity symptoms used in this paper were described by the author in [26]. 

GI involvement was also assessed by the proposed inclusion criteria by Khanna et al. [26] that included the presence of at least one of the following: (1) GI symptoms for at least three of the past seven days—evaluated by UCLA GIT 2.0; (2) abnormal results in GI complementary diagnostic tests; and/or (3) use of pharmacotherapy for the GI system.

The short-form version of the SF-36 health survey (SF-36v2) is a generic instrument designed to measure the self-perception of individuals regarding their health status on a scale from 0 (death) to 100 (perfect health status). [27] It assesses eight dimensions (physical functioning (PF), bodily pain (BP), role limitations due to physical health (RE), general health perception (GH), mental health (MH), role limitations due to emotional problems (RE), vitality (VT), and social functioning (SF)) and provides two component summary measures, one physical (PCS) and the other mental (MCS). In the case of the Portuguese version, [28] these summary measures are normalized to the Portuguese general population.

EQ-5D-5L is a generic preference-based quality of life questionnaire that measures five dimensions (mobility, self-care, usual activities, pain/discomfort, and anxiety/depression). Each dimension has five levels of impairment, allowing us to describe a total of 3125 different health states [29]. A visual analogue scale also asks for self-perception of general health status. Portuguese utilities can be computed by an algorithm based on general public preferences [30] and Portuguese norms are also available [31].

### 2.4. Reliability

We tested the reliability of the Portuguese UCLA GIT 2.0 version through internal consistency, item–total correlation, and intertemporal stability.

Internal consistency was tested by means of the score of the Cronbach’s alpha coefficient, where accepted values should be between 0.70 and 0.90 [18]. Intertemporal stability was tested by the intraclass correlation coefficient (ICC), with two consecutive moments one month apart. We followed the criteria that stipulate that an ICC lower than 0.50 corresponds to a weak correlation; a score between 0.50 and 0.75 and between 0.75 to 0.90, respectively, correspond to a moderate and good correlation; and a score higher than 0.90 corresponds to an excellent correlation [32].

### 2.5. Validity

Content validity was tested during the cultural adaptation we performed on the Portuguese UCLA GIT 2.0 version provided by the author. In addition to this, construct validity tests included both structural validity and hypothesis testing with samples of sociodemographic and clinical variables. Finally, criterion validity was tested by comparing the total GIT score with the scores obtained by the SF-36v2 and EQ-5D-5L [18].

To test structural validity, we performed exploratory factor analysis based on principal components estimates with a previous assessment of the sampling adequacy via the Kaiser–Meyer–Olkin (KMO) indicator and by using Bartlett’s test of sphericity. A KMO smaller than 0.50 or between 0.50 and 0.60 is considered unacceptable or poor, and if between 0.60 and 0.70, between 0.70 and 0.80, between 0.80 and 0.90, or higher than 0.90, is seen, respectively, as fair, average, good, or very good. The significance of the Bartlett sphericity test should be smaller than 0.001 [33].

For the total UCLA GIT score, the hypothesis testing was performed with known sociodemographic (sex, age group) and clinical variables groups, based on the distribution of each variable. Student’s *t*-test was used for two independent variables and ANOVA for more than two independent variables. 

To test the criterion validity, we computed Pearson’s correlations and followed Cohen’s (1988) [34] rule, according to which correlations smaller than 0.30 are considered weak, those between 0.30 and 0.50 are considered moderate, and those higher than 0.50 are considered strong.

The statistical software used was SPSS v28 (IBM, Armonk, NY, USA).

## 3. Results

### 3.1. Sample

Our sample was composed of 102 SSc patients. The sociodemographic and lifestyle variables are presented in Table 1.

All patients approached by the researchers agreed to participate in the study. In general, our sample was composed of females (82.4%) and patients older than 50 years (70.6%), who were married (69.6%), and had at least seven years of education (58.9%). A smaller percentage of them were smokers (5.9%) and/or alcohol drinkers (15.7%).

Table 2 presents the main clinical characteristics measured in this study. 

Of the sample, 62% presented with the lSSc form (vs. diffuse subset in 29.4%) and the majority had fewer than five years of disease duration (56.9%). According to the GI criteria, 59.8% of patients presented with GI involvement associated with the SSc, 82.0% of them showed clinical symptoms, 50.8% were under GI pharmacotherapy, and 55.7% had abnormalities in GI exams. The majority of patients experienced Raynaud’s phenomenon (94.1%) and presented abnormal nailfold capillaries (79.4%). Concerning skin manifestations, sclerodactyly and telangiectasias abnormal nailfold capillaries were the most prevalent (in 63.7% and 62.7% of the patients, respectively), digital tip ulcers were present in 30.4%, and 27.5% presented with puffy fingers and fingertip pitting scars. Lung involvement as interstitial disease or pulmonary arterial hypertension was seen in 21.6% and 8.8% of the patients, respectively.

In addition, 89% of patients had positivity for Antinuclear Antibodies (ANA). Regarding SSc-related autoantibodies, the most common was anticentromere (59%), followed by anti-topoisomerase I (19.6%) and anti-RNA polymerase III (1%).

The health status and quality of life scores of the sample are presented in Table 3. The UCLA GIT 2.0 scores for the sample are shown in Table 4.

High burdens caused by this disease, mainly in the physical health status dimensions, were found.

Table 4 also shows the discrimination power of UCLA GIT 2.0 among the severity levels of patients’ disease.

The means for all the UCLA GIT scores were very low, meaning that the majority of patients had never experienced such GI symptoms. In fact, the observed answers in the various dimensions varied from 34.3% (distension/bloating) to 89.1% (fecal soilage). Contrary to patients without GI symptoms were the domains of reflux and distension/bloating, classified as severe in 15.7%, followed by constipation in 14.7%. The total GIT score was reported as mild in 57.8%, moderate in 16.7%, and severe in only 10.8% of the patients.

Analyzing Table 4, the means for all UCLA GIT dimensions are smaller in asymptomatic patients and increase with the severity of symptoms. In fact, all the UCLA dimensions scores (except fecal soilage) were statistically different between patients with no, mild, moderate, or severe symptoms.

Low UCLA GIT scores were seen in the domains of social functioning and emotional wellbeing, which were not affected in 60.8% and 53.9% of the patients, respectively. Only 13.7% reported severe symptoms relating to emotional wellbeing, compared to 5.9% for social functioning.

### 3.2. Reliability

Table 5 presents the ICC scores from the one-month test–retest on 31 SSc patients. The total GIT score index was highly reliable (ICC = 0.912), with partial scores ranging between good (0.784) and excellent (0.927).

In addition, the item–total correlations for all UCLA GIT 2.0 dimensions ranged from 0.655 to 0.898, very acceptable. The internal consistency Cronbach’s alpha for the total GIT score was 0.954, ranging from 0.672 (diarrhea) to 0.939 (emotional wellbeing).

### 3.3. Validity

Because the Portuguese version of UCLA GIT 2.0 had already been provided by its author, the content validity test was limited to a clinical review with two rheumatologists and a cognitive debriefing with eight patients. Very minor changes were proposed by the experts, and patients did not show any difficulties in understanding the Portuguese version of this measurement instrument. In addition, they did not find any redundancies or ambiguities.

To test structural validity, we used principal component factor analysis, and we were able to find the main dimensions defined by the authors of the original version with 76.3% of the variance explained. The dimensions ‘diarrhea’, ‘emotional wellbeing’, and ‘constipation’ appeared immediately after factor analysis. However, ‘social functioning’, representing the interference of disease with social activities, was divided into two factors, the first one including the dimension ‘soilage’. In addition, the dimensions ‘reflux’ and ‘distension/bloating’ produced two other dimensions, one of them concerning only regurgitating, feeling like vomiting, or vomiting; the other dimension encompassed all the distension items plus heartburn, acid reflux, and sleeping in a raised or seated position. Having difficulty swallowing solid food appeared independently from the other dimensions. The KMO measure of sampling adequacy was 0.855 (good) and the Bartlett’s test of sphericity was associated with a significance smaller than 0.001.

Construct validity also included how the total GIT score and the UCLA GIT 2.0 dimensions behaved when the sample was split on the basis of certain sociodemographic and clinical variables. Table 6 presents the results of the t-Student and ANOVA tests considering the total GIT score and the dimensions that were revealed to be significant.

As we can observe from this table, the total UCLA GIT score is not statistically different when distinct subsamples are considered. However, when the dimension ‘diarrhea’ is analyzed, women showed fewer symptoms than men. Moreover, whenever GI involvement exists, we evidenced significantly higher total UCLA GIT scores as well as in partial domains regarding ‘reflux’, ‘distension/bloating’, and ‘constipation’. Similar conclusions were found when cases of total UCLA GIT equal to zero were excluded.

Lastly, to test the criterion validity of the Portuguese version of UCLA GIT 2.0, we correlated the total GIT scores with the scores from EQ-5D-5L and SF-36v2. The results of these correlation analyses are presented in Table 7.

From this table, we notice a high correlation between any UCLA GIT 2.0 dimension and quality of life (EQ-5D-5L). Regarding self-perception of health status (SF-36v2), both summary scores show, in general, highly significant correlations with the UCLA GIT 2.0 dimensions. There is a natural exception for ‘constipation’, which seems not to be correlated with physical dimensions, namely RP, BP, and GH. We found significant moderate correlations between UCLA GIT ‘emotional wellbeing’ and the SF-36v2 dimensions SF, RE, and MH. Patients with GI involvement presented, although highly significant, weak correlations with EQ-5D-5L and some domains of SF-36v2, namely RP, BP, and MH.

## 4. Discussion

Involvement of the GI tract may be present in 90% of all patients with SSc and it is associated with a decline in HRQoL. Regardless of its high prevalence, GI tract involvement remains poorly investigated in clinical practice, and treatment options are based on symptoms control. With this as a background, the UCLA GIT 2.0 questionnaire enables GI symptom evaluation as a feasible and reliable instrument, and assesses its impact on the HRQoL of SSc patients [4,5,10].

The present study supports prior findings that the Portuguese version of the UCLA GIT 2.0 questionnaire showed very good internal consistency (Cronbach’s alpha = 0.954). The reproducibility measured by the test–retest presented a highly reliable ICC equal to 0.912, and the item–total correlations of the different dimensions were acceptable, ranging from 0.655 to 0.898. These results were similar to those obtained by the author of the original version, as well as by the other language validations [9,11,12,13,14,15,16,25].

The Portuguese version was very well accepted by patients in a cognitive debriefing and approved by the two rheumatologists in content validity.

To test construct validity, we analyzed sociodemographic and clinical criteria, and the total UCLA GIT score was not statistically different in the considered subsamples, except in the diarrhea dimension, where women showed fewer symptoms than men. There is some inconsistency in the literature regarding the influence of sex in GI involvement and severity in SSc patients. McMahan et al. found the male sex strongly associated with severe GI dysmotility [35].

Similarly to other studies, we did not find any difference between disease subset classification, duration, or disease specific autoantibodies and GI involvement [9]. In the French study, patients with lSSc had significantly higher mean scores in the ‘distention/bloating’ (1.16 vs. 0.72), ‘constipation’ (0.48 vs. 0.27), and total GIT (0.65 vs. 0.38) domains [11]. The Serbian study was the only one reporting a higher mean distension scale score in anti-centromere patients compared with anti-topoisomerase I positive patients (1.0 vs. 0.56, *p* = 0.05) and, excluding the ‘diarrhea’ domain, all others mean scores were also higher in anti-centromere positive patients, although not reaching statistical significance [16].

Concerning GI involvement, patients with symptoms, under GI therapy, or showing alterations in GI complementary testing presented significantly higher total UCLA GIT scores and in partial domain scores regarding ‘reflux’, ‘distension/bloating’, and ‘constipation’. In the Dutch study, there was a significant difference of all subscales of the UCLA GIT, except for ‘constipation’, between patients with and without GI diagnoses [12].

Concerning criterion validity, we found a high negative correlation between any UCLA GIT 2.0 dimension and quality of life (EQ-5D-5L), the strongest correlation being between emotional wellbeing (r^2^ = −0.532 for index and −0.439 for VAS) and total UCLA GIT score (r2 = −0.475 for index and −0.463 for VAS). Patients with GI involvement also showed a significant correlation with EQ-5D-5L, although weaker (r2 = −0.198 for index and −0.224 for VAS).

In previous validations, only the Italian one included the EQ-5D questionnaire, and the results were in line with our study, reporting that UCLA GI 2.0 emotional wellbeing significantly correlated more with the impact on usual activities and to a lesser extent with the limitation of physical problems [14].

In the SF-36v2 questionnaire, both summary scores showed, in general, highly significant correlations with UCLA GIT 2.0 dimensions, exception for ‘constipation’, which seems not to be correlated with physical dimensions, namely RP, BP, and GH. 

The strongest correlations between UCLA GIT 2.0 ‘emotional wellbeing’ and the SF-36v2 were in the GH, SF, RE, and MH domains (r2 between −0.406 and −0.472). The French, Dutch, Italian, Romanian, Chinese, and Turkish validations also reported weak to moderate correlation between UCLA GIT 2.0 ‘emotional wellbeing’ and some domains of SF-36v2, namely RE and MH, but not with GH or SF [11,12,13,14,15,25].

The social functioning domains of UCLA GIT 2.0 and SF-36v2 presented a negatively moderate correlation of −0.368, in line with the Italian, Singapore and Romanian validations. [13,14,15] On the other hand, in the French and Turkish validations, although the social functioning scales between the two instruments had a statistical association, the correlation was weak (r2 = −0.27, *p* < 0.05) [11,25].

Contrary to MCS, which showed highly significant correlations with all UCLA GIT 2.0 dimensions, PCS was not correlated with either the ‘fecal soilage’ or ‘social function’ UCLA GIT 2.0 dimensions. Moderate correlations were seen between PCS and ‘reflux’ and total GIT score, whereas weak correlations were found between PCS and ‘constipation’, ‘distension/bloating’, ‘diarrhea’, and ‘emotional wellbeing’, regarding the UCLA GIT 2.0 instrument. In the Turkish validation, a moderate negative correlation was found between the SF-36 PCS and the ‘reflux’, ‘distension’, ‘diarrhea’, ‘social functioning’, ‘emotional wellbeing’, and total UCLA scores [25]. In the French study, a moderate correlation was also found between the SF-36 PCS and the ‘reflux’, ‘distension’, ‘social functioning’, and total UCLA scores [11]. Similarly, a moderate negative correlation was found between the SF-36v2 PCS and the ‘distension’ and total UCLA scores in the Dutch study only [12].

We found a significant moderate correlation between the physical function of SF-36v2 and all symptom domains of the UCLA GIT 2.0, including the total UCLA GIT score, similarly to the Turkish validation [25]. No correlation was found between the UCLA GIT 2.0 domains and the physical functioning subscales in the French, Dutch, and Romanian studies, or in the Italian validation, where there was no significant correlation between these subscales, except for ‘constipation’ and RP of SF-36v2 [11,12,13,14]. 

Total GIT score was the domain where correlations were stronger and significantly higher for the mental health scales than for the PCS score of the SF-36. These results are similar to other validations [9,11,11,12,13,15,16,25].

Patients with GI involvement presented, although highly significant, weak correlations with some domains of SF-36v2, namely RP, BP, and MH.

Different results from other validation papers may be explained by the diversity of SSc patients (sample sizes and clinical characteristics) and cultural differences of our sample when compared with their studies.

### Limitations and Future Directions

Although we have complied with the minimum same size to validate a measurement instrument, we consider that it would be advantageous to replicate the study with a larger sample. This would probably provide us with better variability of the analyzed variables.

## 5. Conclusions

Strictly following methodological criteria, we have created and validated a Portuguese version of UCLA GIT 2.0. This version of UCLA GIT 2.0 showed good psychometric properties and can be used in research and clinical practice for the assessment of GI involvement in SSc patients.

## Figures and Tables

**Table 1 ijerph-20-01553-t001:** Sociodemographic characteristics of patients (*n* = 102).

Variable	Value	No.	%
Sex	Female	84	82.4
	Male	18	17.6
Age	<50	30	29.4
	[50–65)	41	40.2
	≥65	31	30.4
	Minimum–Maximum	27–83	
	Median	56.5	
	Mean ± standard deviation	57.0 ± 12.5	
Marital status	Single	14	13.7
	Married/living with a partner	71	69.6
	Divorced/separated	10	9.8
	Widowed	7	6.9
Employment status	Employed	51	50.0
	Not employed	10	9.8
	retired	37	36.3
	housekeeper	4	3.9
Years of education	No formal education	2	2.0
	4 years	29	28.4
	5–6 years	11	10.8
	7–9 years	17	16.7
	10–12 years	32	31.4
	More than 12 years	11	10.8
Lifestyle	Smoker	6	5.9
	Alcohol drinker	16	15.7

**Table 2 ijerph-20-01553-t002:** Disease characteristics of patients (*n* = 102).

Variable	Value	*n*	%
Disease duration since diagnosis (years)	[0–5)	58	56.9
	≥5	44	43.1
	Minimum–Maximum	0–37	
	Median	4	
	Mean ± standard deviation	5.6 ± 6.0	
Subset classification of SSc	Diffuse	30	29.4
	Limited	64	62.7
	VEDOSS	7	6.9
	Sine scleroderma	1	1.0
2013 ACR/EULAR classification criteria	Skin thickening of the fingers of both hands	48	47.1
	Puffy fingers	28	27.5
	Sclerodactyly	65	63.7
	Digital tip ulcers	31	30.4
	Fingertip pitting scars	28	27.5
	Telangiectasia	64	62.7
	Abnormal nailfold capillary	81	79.4
	Pulmonary arterial hypertension	9	8.8
	Interstitial lung disease	22	21.6
	Raynaud’s phenomenon	96	94.1
Autoantibodies	ANA	91	89.2
	Anti-centromere	59	59
	Anti-topoisomerase I	20	19.6
	Anti-RNA polymerase III	1	1.0
	Other antibodies	13	12.7
GI involvement	Yes	61	59.8
	Clinical symptoms ^1,2^	50	82.0
	GI pharmacotherapy ^2^	31	50.8
	Abnormal GI exams ^2^	34	55.7

GI: gastrointestinal; VEDOSS: very early diagnosis of systemic sclerosis; ANA: Antinuclear Antibodies; ^1^ with GI; ^2^ The percentage was adjusted to the subsample of patients with GI involvement.

**Table 3 ijerph-20-01553-t003:** Health status and quality of life variables.

		Min	Max	Mean	Sd
SF-36v2	Physical functioning (PF)	5.0	100.0	61.9	24.4
	Physical role functioning (RP)	0.0	100.0	57.0	31.4
	Bodily pain (BP)	0.0	100.0	49.7	22.3
	General health perceptions (GH)	0.0	87.0	41.4	19.4
	Vitality (VT)	10.0	100.0	43.7	24.1
	Social role functioning (SF)	0.0	100.0	70.1	28.6
	Emotional role functioning (RE)	0.0	100.0	64.3	29.4
	Mental health (MH)	4.0	100.0	60.1	27.7
	Physical component summary (PCS)	13.8	62.4	39.3	10.3
	Mental component summary (MCS)	22.0	70.2	47.5	12.1
EQ-5D-5L	Index	0.08	1.00	0.77	0.20
	VAS	4.0	100.0	66.8	18.7

Sd: standard deviation; VAS: visual analogic scale.

**Table 4 ijerph-20-01553-t004:** Means of UCLA GIT 2.0 for different GI severity symptoms.

	All Sample (*n* = 102)	No Symptoms (*n* = 49)		Mild Symptoms (*n* = 13)		Moderate Symptoms (*n* = 25)		Severe Symptoms (*n* = 15)			
	Mean	Mean	%	Mean	%	Mean	%	Mean	%	F	Sig.
Reflux	0.46	0.29	38.2	0.40	19.6	0.53	26.5	0.94	15.7	6.800	<0.001
Distension/bloating	0.69	0.36	34.3	0.61	41.2	1.04	8.8	1.25	15.7	8.976	<0.001
Fecal soilage	0.16	0.12	89.1	0.08	8.9	0.24	2.0	0.20	0.0	0.482	0.695
Diarrhea	0.36	0.20	59.8	0.58	0.0	0.52	31.4	0.43	8.8	2.991	0.035
Social functioning	0.29	0.10	60.8	0.29	13.7	0.40	19.6	0.69	5.9	7.065	<0.001
Emotional wellbeing	0.37	0.13	53.9	0.19	20.6	0.60	11.8	0.89	13.7	8.882	<0.001
Constipation	0.47	0.00	48.0	0.25	12.7	0.75	24.5	1.75	14.7	320.246	<0.001
Total GIT score	0.39	0.20	14.7	0.36	57.8	0.56	16.7	0.73	10.8	8.164	<0.001

F: Fisher’s F for comparison between no symptoms, mild, moderate or severe; Sig: *p*-value.

**Table 5 ijerph-20-01553-t005:** UCLA GIT 2.0 reliability scores.

UCLA Items	Number of Items	Item-Total Correlation	Cronbach’s Alpha	ICC	ICC 95% IC
Reflux	8	0.778	0.829	0.924	0.843–0.964
Distension/bloating	4	0.876	0.837	0.927	0.848–0.965
Fecal soilage	1	-	-	0.784	0.552–0.896
Diarrhea	2	0.655	0.672	0.674	0.324–0.843
Social functioning	6	0.849	0.865	0.851	0.691–0.928
Emotional well-being	9	0.898	0.939	0.853	0.696–0.929
Constipation	4	-	0.841	0.769	0.520–0.888
Total GIT score	34	-	0.954	0.912	0.817–0.958

**Table 6 ijerph-20-01553-t006:** Total UCLA GIT scores for different levels of sociodemographic and clinical variables.

			*n*	Mean	SD	|*t*| or F	Sig
Sex	Total UCLA-GIT	Female	84	0.37	0.47	1.081	0.289
		Male	18	0.48	0.39		
	Diarrhea	Female	84	0.27	0.46	3.783	<0.001
		Male	18	0.78	0.71		
Age	Total UCLA-GIT	Less than 50	30	0.33	0.39	0.607	0.547
		(50–65)	41	0.45	0.45		
		65 or more	31	0.36	0.36		
Disease duration (years)	Total UCLA-GIT	(0–5)	58	0.36	0.39	0.776	0.440
		5 or more	44	0.43	0.53		
Subset classification of SSc	Total UCLA-GIT	Diffuse	30	0.40	0.39	0.245	0.807
		Not diffuse	72	0.38	0.48		
Sclerodactyly	Total UCLA-GIT	No	37	0.38	0.45	0.043	0.966
		Yes	65	0.39	0.48		
Digital tip ulcers	Total UCLA-GIT	No	71	0.37	0.44	0.517	0.607
		Yes	31	0.43	0.50		
Telangiectasia	Total UCLA-GIT	No	38	0.34	0.45	0.844	0.401
		Yes	64	0.42	0.46		
Anti-centromere	Total UCLA-GIT	No	41	0.32	0.37	1.317	0.191
		Yes	59	0.44	0.51		
GI involvement	Total UCLA-GIT	No	41	0.26	0.46	2.419	0.016
		Yes	61	0.48	0.43		
	Reflux	No	41	0.24	0.37	3.529	<0.001
		Yes	61	0.60	0.58		
	Distension	No	41	0.43	0.67	2.876	0.002
		Yes	61	0.86	0.80		
	Constipation	No	41	0.27	0.45	2.628	0.005
		Yes	61	0.61	0.73		

SD: standard deviation; *t*: *t*-Student value; F: ANOVA value; Sig: *p*-value.

**Table 7 ijerph-20-01553-t007:** Spearman’s correlation coefficients between UCLA GIT 2.0 and the EQ-5D-5L and SF-36v2 scales.

		EQ-5D-5L		SF-36v2									
UCLA-GIT 2.0		Index	VAS	PCS	MCS	PF	RP	BP	GH	VT	SF	RE	MH
Reflux	r	−0.302	−0.323	−0.368	−0.289	−0.359	−0.342	−0.444	−0.381	−0.429	−0.359	−0.312	−0.310
	sig	0.002	0.001	<0.001	0.003	<0.001	<0.001	<0.001	<0.001	<0.001	<0.001	0.001	0.002
Distension/bloating	r	−0.367	−0.331	−0.263	−0.359	−0.308	−0.226	−0.400	−0.402	−0.384	−0.426	−0.310	−0.349
	sig	<0.001	0.001	0.008	<0.001	0.002	0.022	<0.001	<0.001	<0.001	<0.001	0.002	<0.001
Fecal soilage	r	−0.359	−0.346	−0.180	−0.295	−0.227	−0.298	−0.174	−0.249	−0.242	−0.257	−0.358	−0.291
	sig	<0.001	<0.001	0.070	0.003	0.022	0.002	0.080	0.012	0.014	0.009	<0.001	0.003
Diarrhea	r	−0.290	−0.340	−0.213	−0.278	−0.282	−0.252	−0.293	−0.246	−0.256	−0.329	−0.280	−0.269
	sig	0.003	<0.001	0.031	0.005	0.004	0.011	0.003	0.013	0.009	0.001	0.004	0.006
Social functioning	r	−0.374	−0.417	−0.153	−0.387	−0.243	−0.272	−0.297	−0.244	−0.272	−0.368	−0.382	−0.360
	sig	<0.001	<0.001	0.123	<0.001	0.014	0.006	0.002	0.013	0.006	<0.001	<0.001	<0.001
Emotional wellbeing	r	−0.532	−0.439	−0.265	−0.444	−0.370	−0.348	−0.328	−0.406	−0.393	−0.472	−0.424	−0.433
	sig	<0.001	<0.001	0.007	<0.001	<0.001	<0.001	0.001	<0.001	<0.001	<0.001	<0.001	<0.001
Constipation	r	−0.356	−0.279	−0.109	−0.325	−0.236	−0.195	−0.192	−0.186	−0.260	−0.338	−0.289	−0.269
	sig	<0.001	0.004	0.276	0.001	0.017	0.050	0.054	0.062	0.008	0.001	0.003	0.006
Total GIT score	r	−0.475	−0.463	−0.310	−0.439	−0.384	−0.364	−0.420	−0.420	−0.427	−0.478	−0.435	−0.430
	sig	<0.001	<0.001	0.001	<0.001	<0.001	<0.001	<0.001	<0.001	<0.001	<0.001	<0.001	<0.001
GI involvement	r	−0.198	−0.224	−0.179	−0.126	−0.174	−0.205	−0.250	−0.156	−0.144	−0.185	−0.055	−0.201
	sig	0.046	0.024	0.072	0.207	0.081	0.039	0.011	0.119	0.150	0.063	0.583	0.043

r: Pearson correlation coefficient; sig: *p*-value; PF: physical functioning; RP: physical role functioning; BP: bodily pain; GH: general health perceptions; VT: vitality; SF: social role functioning; RE: emotional role functioning; MH: mental health; PCS: physical component summary; MSM: mental component summary.

## Data Availability

The data that support the findings of this study are available from the corresponding author upon reasonable request.

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
