# Peer review of "Creation and Validation of a Portuguese Version of the UCLA Scleroderma Clinical Trial Consortium Gastrointestinal Tract Instrument"

_ijerph, 2023, doi:10.3390/ijerph20021553_

Round 1
Reviewer 1 Report
In this manuscript the authors present the validation of a Portuguese version of the UCLA Scleroderma Clinical Trial Consortium GI instrument (UCLA-GIT-2.0). The authors have rigorously validated the version of the questionnaire, the methods used are sound and the results are important. I would like to congratulate the authors for all the effort in this nice piece of work. Nevertheless, there are some minor aspects to be pointed out:1.Table 1 misses the head of table (n and %)
2.Some minor spelling errors in table 1 and 2: when defining an interval, the brackets should be [ ] or [ and ) if the last value of the interval is not included but not [ and [. Also, median value would be important to be presented when reporting min and max value. 3.Table 4, were there any statistically significant differences in scores between patients with no symptoms, mild, moderate or severe? 4.Table 6 the p values are after performing post-hoc corrections? It should be mentioned in text and material and method section what post-hoc tests were used. 5. The statistical software that was used should be presented in the methods section. 6. Maybe the authors could upload the final version of the questionnaire as a supplementary table? Best regards!Author Response
Dear Reviewer,
The authors are thankful for the comments and suggestions pointed that we agree will improve this manuscript. Next, we will focus on your comments.
In this manuscript the authors present the validation of a Portuguese version of the UCLA Scleroderma Clinical Trial Consortium GI instrument (UCLA-GIT-2.0). The authors have rigorously validated the version of the questionnaire, the methods used are sound and the results are important. I would like to congratulate the authors for all the effort in this nice piece of work. Nevertheless, there are some minor aspects to be pointed out:
Thank you for your comment.
- Table 1 misses the head of table (n and %)
The heading is, in fact, missing. It was deleted by accident. The problem is already solved, thank you.
- Some minor spelling errors in table 1 and 2: when defining an interval, the brackets should be [ ] or [ and ) if the last value of the interval is not included but not [ and [.
Also, median value would be important to be presented when reporting min and max value.
We changed table layout according to the journal directions.
- Table 4, were there any statistically significant differences in scores between patients with no symptoms, mild, moderate or severe?
In fact, all the UCLA dimensions scores (except fecal soilage) were statistically different between patients with no symptoms, mild, moderate or severe. We have included this last sentence in the text, as well as two extra columns in table 4 with the statistical significance of these differences.
- Table 6 the p values are after performing post-hoc corrections? It should be mentioned in text and material and method section what post-hoc tests were used.
No, there were no post-hoc corrections.
- The statistical software that was used should be presented in the methods section.
The statistical software used was the SPSS v.28. We have included this information at the end of the Methods section.
- Maybe the authors could upload the final version of the questionnaire as a supplementary table?
Unfortunately, we are not allowed to place the Portuguese version of the UCLA-GIT 2.0 in the paper. The reason is because all the versions of a health measurement instrument belong to the original authors. We have only asked him permission to translate and validate the UCLA-GIT 2.0 in European Portuguese. We do not have any rights over the measure, including the Portuguese version.
Reviewer 2 Report
The UCLA GIT 2.0 questionnaire is a reliable instrument to evaluate GI symptoms in patients with SSc. A Portuguese version was already available, but it was not formally validated. A new Portuguese version with “very minor changes” was administered to 102 patients to validate it.
I would suggest to make the final version of the questionnaire available as a supplementary file.
Table 4 and the paragraph commenting it in the results should be clarified. If I understand properly, the mean GIT scores observed in patients with different GI symptoms severity are reported. However, it is not clear (at least, for me) how these groups were defined: was there an external anchor available? (eg self-rated severity of GIT? If so, this should be indicated in the Methods; if not, was it a circular correlation with the UCLA questionnaire itself? is it sound?).
Unfortunately, the paragraph commenting table 4 did not help me to understand this analysis, and, in my opinion, it should be rephrased.
By the way, I noticed that while all the other tables titles are purely descriptive (eg Table 6. Total UCLA-GIT scores for different levels of sociodemographic and clinical variables), Table 4 title is instead an assertive comment of the results (Means of UCLA-GIT 2.0 differentiate between severity).
A large part of the Discussion simply consists in a brief presentation of the results of previous similar studies who validated UCLA-GIT in different languages. Some discrepant results were sometimes observed (eg different correlations between GIT subscales and other PROs), but the authors did not try to discuss them. Indeed, these discrepancies might be casual observations due to the small samples evaluated in each of these studies and the large numbers of comparisons or correlations calculated. I would suggest to shorten this section of the manuscript accounting for this possible explanation.
There are some inaccuracies (eg Table 1 legend refers rather to Table 2), suggesting the need of a more careful revision of the manuscript. Editing of English language and style is also recommended.
Minor points
- I would suggest to report how many consecutive patients did not participate to this study (eg not-Portuguese speaking; denied content, etc) and whether any missing data was recorded.
- in Table 1 data on lifestyles are reported; their impact on the questionnaire, however, was not reported. Anyway, if presented, they should be better defined. eg: smokers (current/ever?) drink (definition?)
- since data on GI drugs were collected I would suggest to present them (eg in Table 2)
- page 5, line 186: data on UCLA-GIT are said to be presented in Table 3, which is not true.
- data on EQ- 5D-5L were collected. Since the correlations of UCLA-GIT with Index and VAS are presented in table 7, I suggest to briefly describe in the Methods how EQ- 5D Index and VAS were calculated. Is it worth to present separate data on correlations with the 5 EQ-5D dimensions?
- ICC was tested on 31 (randomly selected) patients. Did any patient develop new GI symptoms in this time interval? Were such patients excluded from retesting?
Author Response
Dear Reviewer,
The authors are thankful for the comments and suggestions pointed that we agree will improve this manuscript. Next, we will focus on your comments.
- The UCLA GIT 2.0 questionnaire is a reliable instrument to evaluate GI symptoms in patients with SSc. A Portuguese version was already available, but it was not formally validated. A new Portuguese version with “very minor changes” was administered to 102 patients to validate it. I would suggest to make the final version of the questionnaire available as a supplementary file.
Unfortunately, we are not allowed to place the Portuguese version of the UCLA-GIT 2.0 in the paper. The reason is because all the versions of a health measurement instrument belong to the original authors. We have only asked him permission to translate and validate the UCLA-GIT 2.0 in European Portuguese. We do not have any rights over the measure, including the Portuguese version.
- Table 4 and the paragraph commenting it in the results should be clarified. If I understand properly, the mean GIT scores observed in patients with different GI symptoms severity are reported. However, it is not clear (at least, for me) how these groups were defined: was there an external anchor available? (eg self-rated severity of GIT? If so, this should be indicated in the Methods; if not, was it a circular correlation with the UCLA questionnaire itself? is it sound?).
Unfortunately, the paragraph commenting table 4 did not help me to understand this analysis, and, in my opinion, it should be rephrased.
We have included the text “The levels of GI severity symptoms used in this paper were described by the author in [26]” to explain the cutoff points to determine the severity of GI symptoms.
- By the way, I noticed that while all the other tables titles are purely descriptive (eg Table 6. Total UCLA-GIT scores for different levels of sociodemographic and clinical variables), Table 4 title is instead an assertive comment of the results (Means of UCLA-GIT 2.0 differentiate between severity).
We changed the title for table 4 to “Means of UCLA-GIT 2.0 for different GI severity symptoms”.
- A large part of the Discussion simply consists in a brief presentation of the results of previous similar studies who validated UCLA-GIT in different languages. Some discrepant results were sometimes observed (eg different correlations between GIT subscales and other PROs), but the authors did not try to discuss them. Indeed, these discrepancies might be casual observations due to the small samples evaluated in each of these studies and the large numbers of comparisons or correlations calculated. I would suggest to shorten this section of the manuscript accounting for this possible explanation.
We added the following sentence at the end of the Discussion: “Different results from other validation papers may be explained by the diversity of SSc patients (sample sizes and clinical characteristics) and cultural differences of our sample when compared with their studies.” We haven’t explained the discrepancies just after any comparison to avoid repetitions in the text, which would substantially increase its length.
- There are some inaccuracies (eg Table 1 legend refers rather to Table 2), suggesting the need of a more careful revision of the manuscript.
Thank you for the comment. We have not noticed it.
- Editing of English language and style is also recommended.
After your comment we have revised the manuscript. The paper was revised in linguistic terms by a native English translator
- I would suggest to report how many consecutive patients did not participate to this study (eg not-Portuguese speaking; denied content, etc) and whether any missing data was recorded.
In the beginning of the sample description we included the following sentence “All patients addressed by the researchers accepted to participle in the study”. Therefore, we did not find any missing data.
- in Table 1 data on lifestyles are reported; their impact on the questionnaire, however, was not reported. Anyway, if presented, they should be better defined. eg: smokers (current/ever?) drink (definition?)
We have not included the lifestyle variables in the data analysis just because the non-smokers and/or non-alcohol drinkers were the large majority of our sample. For instance, for the smokers the split was between 6 smokers compared to 96 non-smokers and all comparisons could run the risk of being biased.
- since data on GI drugs were collected I would suggest to present them (eg in Table 2)
As this is a validation paper, we considered that this detail was not needed. However, to turn the text cleared we added in table 2, the reference of GI involvement patients with clinical symptoms, pharmacotherapy and abnormal GI exams. Therefore, we included the following sentence after table 2: “82.0% of them showed clinical symptoms, 50.8% were under GI pharmacotherapy, and 55.7% had abnormalities in GI exams”.
- page 5, line 186: data on UCLA-GIT are said to be presented in Table 3, which is not true.
You are absolutely right. The main descriptive scores of the Portuguese version of UCLA-GIT are only presented in table 4 under the heading ‘All sample (n=102)’. Therefore, in the previous line 186, we rephrased the sentence to “Health status and quality of life scores given by this sample are presented in table 3. UCLA-GIT 2.0 scores for all sample are shown in table 4”.
- data on EQ- 5D-5L were collected. Since the correlations of UCLA-GIT with Index and VAS are presented in table 7, I suggest to briefly describe in the Methods how EQ- 5D Index and VAS were calculated. Is it worth to present separate data on correlations with the 5 EQ-5D dimensions?
Thank you for your comment.
At the end of ‘Measurement instruments’ heading from ‘Material and Methods’ section, we describe the EQ-5D-5L. We have added a reference to the VAS with the following text: “A visual analogue scale also asks for self-perception of general health status.
However, in what concerns how a utility score is computed from the descriptive system with the five dimensions, we included the reference 30 published in Quality of Life Research journal.
- ICC was tested on 31 (randomly selected) patients. Did any patient develop new GI symptoms in this time interval? Were such patients excluded from retesting?
We were sure that no substantial change in health status occurred in patients during the test-retest. No patient was excluded by this reason.
Round 2
Reviewer 2 Report
No further comment.